# Fertility Preservation in Children and Adolescents during Oncological Treatment—A Review of Healthcare System Factors and Attitudes of Patients and Their Caregivers

**DOI:** 10.3390/cancers15174393

**Published:** 2023-09-02

**Authors:** Piotr Pawłowski, Karolina Joanna Ziętara, Justyna Michalczyk, Magdalena Fryze, Anna Buchacz, Agnieszka Zaucha-Prażmo, Joanna Zawitkowska, Anna Torres, Marzena Samardakiewicz

**Affiliations:** 1Student Scientific Association at the Department of Psychology, Faculty of Medicine, Medical University of Lublin, 20-093 Lublin, Poland; pawlowskipiotr56@gmail.com (P.P.); justynaelwiramichalczyk@gmail.com (J.M.); 2Department of Psychology, Psychosocial Aspects of Medicine, Medical University of Lublin, 20-093 Lublin, Poland; magdalenafryze@umlub.pl (M.F.); marzena.samardakiewicz@umlub.pl (M.S.); 3Youth Cancer Europe, 400372 Cluj-Napoca, Romania; ania@fundacjapaniani.org; 4Department of Pediatric Hematology, Oncology and Transplantology, Medical University of Lublin, 20-093 Lublin, Poland; agnieszka.zaucha-prazmo@umlub.pl (A.Z.-P.); joanna.zawitkowska@umlub.pl (J.Z.); 5Department of Pediatric and Adolescent Gynecology, Medical University of Lublin, 20-093 Lublin, Poland; anna.torres@umlub.pl

**Keywords:** oncofertility, fertility preservation, adolescent, children

## Abstract

**Simple Summary:**

Oncofertility refers to medical interventions aimed at preserving the fertility of cancer patients, particularly those undergoing treatments like chemotherapy and radiation therapy that can harm reproductive cells. This literature review focuses on oncofertility in pediatric and adolescent populations, a relatively niche area with limited research compared to adults. The review examines the methods used, financing, ethical considerations, and the perspectives of patients and their parents. In prepubertal patients, there are fewer fertility preservation options available compared to pubertal individuals. The funding for these procedures varies by country, with only a few governments choosing to provide reimbursement. Oncofertility in pediatric and adolescent patients raises controversies, including decisions, parental beliefs, partner considerations, ethical dilemmas, and healthcare professionals’ knowledge and experience. Given the fertility risks young cancer patients face, healthcare professionals must make every effort to help them fulfill their future reproductive plans and desires for a family. The development of systemic solutions is crucial to advance oncofertility in pediatric and adolescent populations.

**Abstract:**

Oncofertility is any therapeutic intervention to safeguard the fertility of cancer patients. Anti-cancer therapies (chemotherapy, radiation therapy, etc.) entail the risk of reproductive disorders through cytotoxic effects on gamete-building cells, especially those not yet fully developed. This literature review analyzes the available data on securing fertility in pediatric and adolescent populations to identify the methods used and describe aspects related to financing, ethics, and the perspective of patients and their parents. Topics related to oncofertility in this age group are relatively niche, with few peer-reviewed articles available and published studies mostly on adults. Compared to pubertal individuals, a limited number of fertility preservation methods are used for prepubertal patients. Funding for the procedures described varies from country to country, but only a few governments choose to reimburse them. Oncofertility of pediatric and adolescent patients raises many controversies related to the decision, parents’ beliefs, having a partner, ethics, as well as the knowledge and experience of healthcare professionals. As the fertility of young cancer patients is at risk, healthcare professionals should make every effort to provide them with an opportunity to fulfill their future reproductive plans and to have a family and offspring. Systemic solutions should form the basis for the development of oncofertility in pediatric and adolescent populations.

## 1. Introduction

Cancer in the 21st century has become a major challenge to medicine. The number of oncological diagnoses, including those affecting the pediatric population, is increasing every year. Childhood cancer is also one of the most common causes of death. Among pediatric and adolescent patients worldwide, leukemias, brain and central nervous system tumors, lymphomas, and sarcomas are the most common cancers. It should be emphasized that in recent years, the rapid development of medicine has led to some significant improvements in survival rates [1,2,3,4,5,6].

Improving survival rates translates into an increase in the percentage of patients interested in having genetic offspring. Available research data show the following correlation: being diagnosed with cancer in childhood increases survivors’ motivation to have offspring in the future [7,8,9,10]. However, the very same systemic treatment or radiation that saves lives has a negative impact on the patient’s future fertility because of the gonadotoxic effects in males and the depletion of ovarian follicles, significantly affecting females. Therefore, when planning treatment for pediatric and adolescent patients, the opportunity to preserve their fertility should be a standard protocol [11,12,13,14,15]. 

Oncofertility care includes fertility preservation (FP)—a broad term that encompasses not only the direct process of preserving fertility in cancer patients but also any secondary disorders such as hormone homeostasis disorders, irregular periods, bleeding, sexual dysfunction, and psychosexual support [16,17,18,19,20,21]. Fertility is an integral part of the human psyche, an often overlooked issue due to the prioritization of biological aspects [22,23,24]. It is indicated as one of the top 5 unmet needs among pediatric and adolescent patients diagnosed with cancer [22,25]. Treatment-induced gonadal dysfunction can eventually lead to complete infertility, making it impossible to have offspring. This complication is irreversible, while others, such as chronic fatigue, baldness, and gastrointestinal disorders, are transient [13,26,27,28,29,30,31]. The negative impact on the future of reproductive plans is also compounded by the fear and anxiety of the patients and their families, which greatly affects their quality of life [32,33,34]. Aspects of future fertility are often overlooked in medical practice due to their delay in time and insufficient understanding of the problem by some medical professionals. Their appearance years later advocates the benefit of early counseling and interventions to preserve fertility in pediatric and adolescent patients, especially in high-risk groups. Providing patients with high-quality care in the future guarantees a functionally optimal biopsychosocial model of wellness [34,35,36,37]. Currently, many methods of preserving fertility are still experimental, although they increasingly and systematically obtain national healthcare providers’ (HCP) approval [38,39,40,41,42]. Many countries around the world have yet to establish regulations and guidelines for safeguarding fertility in pediatric and adolescent patients. Poland is also in this group. Although a working group has already been established at the Polish Society of Gynecologic Oncology, the published recommendations do not differentiate between adult and pediatric populations [38,39,40].

The main purpose of this study is to analyze FP in children treated with systemic oncology, considering the perspective of healthcare systems, the situation of patients, their caregivers and healthcare professionals with regard to FP availability within exemplary healthcare systems, barriers to care, and as well as attitudes of the patients themselves and their caregivers. This study is also intended to draw the medical community’s attention to the need for education, counseling, and implementation of FP methods in pediatric and adolescent patients.

## 2. Materials and Methods

The study was based on a nonsystematic (purposive) review of English-language literature. All details of the included research material are shown in Figure 1. The time limit was set as years 2005–2022. The following databases were searched: PubMed, Scopus, Web of Science, and Google Scholar. The main research question was as follows: What are the options for securing fertility in adolescent and pediatric patients, considering the perspective of parents and cancer patients? The keywords were selected accordingly: Pediatric OR adolescents OR prepubertal; Neoplasms OR cancer OR oncology; ‘Fertility preservation’ OR oncofertility; ‘Reproductive care’; Communication; Psychosocial; Ethics. The results of the database searches were initially analyzed by title and abstract. In the next stage of the research process, two reviewers independently reviewed the full content of the accepted articles. In the event of a difference in the decision as to whether a particular publication should be included or not, a discussion was held among all members of the research group until a consensus was reached. The above-described action was aimed at ensuring a high level of credibility. The analyzed acts, standards, and recommendations included in the analysis came from government sites, health system organizations, non-government organizations, and networks of international associations.

## 3. Results

### 3.1. Fertility Preservation Methods: To Whom, When and How They Should Be Offered

The main method of preserving fertility in male patients is sperm cryopreservation, which should be performed before the initiation of chemotherapy. In pubertal males, the most efficient method of sperm retrieval is by masturbation, which can be offered to patients with Tanner stage 2 (testicular volume of ≥4 cc) [41,42] or Tanner stage 3 (testicular volume of ≥6 cc) [42], depending on the recommendations. Good-quality semen, possible to collect with masturbation, was found in 50% of boys aged 14, with the average volume of the testicles of 8–15 mL [43,44,45,46]. If this cannot be achieved, penile vibration, electro-ejaculation or testicular sperm extraction are alternatives to obtain mature sperm. Testicular tissue cryopreservation is offered to prepubertal patients with a high risk of gonadal failure (high-dose alkylating agents, radiotherapy to the testes or HSCT) in some centers as part of clinical trials. However, it is still considered an investigational procedure, and successful maturation of the sperm from a cryopreserved tissue has not yet been described in humans [17,47,48,49,50]. Other methods, which are also considered experimental, include autologous testicular tissue grafting, spermatogonia stem cell transplantation, and in vitro spermatogenesis [48,51,52]. The first reports on the generation of reproductive cells from immature testicular tissue were also published [53,54].

Methods of FP in female patients include embryo cryopreservation, mature oocyte cryopreservation (OC), and ovarian tissue cryopreservation (OTC). In vitro maturation (IVM) of oocytes was also described when OC or OTC was performed, although some authors argue against this method, especially in very young patients [55,56,57,58]. Therefore, it is still considered experimental. The shielding of the ovarian field and transposition are also utilized in the case of radiotherapy administered to the pelvis [12,27].

Gonadotropin-releasing hormone (GnRH) analogs have been considered a pharmacological method of FP, but there is insufficient evidence to support this thesis. Nevertheless, GnRH analogs are still commonly used in cancer patients for the prevention of abnormal uterine bleeding during cancer treatment [59,60,61].

Embryo cryopreservation and OC have established methods of FP in adult women [62,63]. They were also found suitable for post-pubertal children and adolescents. Both techniques were extensively described in the relevant literature, and various modifications to improve pregnancy outcomes were proposed [64,65,66,67,68,69]. With regard to pediatric patients treated for cancer, there are, however, certain specific issues that need to be considered. Firstly, the cryopreservation of oocytes and embryos can be performed only after puberty has occurred. Secondly, to minimize the delay in anticancer therapy, the random start ovarian stimulation protocol is usually utilized, which has resulted in obtaining the accepted number of good-quality oocytes compared to traditional protocols [59,60,65,70,71,72,73,74]. Thirdly, oocyte retrieval may require transvaginal ovarian puncture, which may be perceived as invasive by some patients and caregivers, thus requiring meticulous counseling with an explanation that hymen restoration can be performed afterwards. Finally, although embryo freezing produces the best rate of subsequent pregnancies, this method may present some difficult psychological, ethical, and legal issues for adolescents [17,75]. 

Ovarian tissue cryopreservation (OTC) is the method of choice for prepubertal patients and for those who cannot succumb to delays in therapy. It involves a laparoscopic surgical retrieval of a part or the whole ovary (in younger patients) and can often be performed at the time of other procedures requiring general anesthesia. Meticulous laboratory workup of the ovarian tissue before freezing and strict quality control of all procedures is crucial to obtaining good pregnancy outcomes, as has been described by several research groups to date [17,60,76,77,78]. Based on the most recent data, OTC and OTT are safe and efficient methods of FP, and they are no longer considered experimental. 

The return of physiological function was achieved in more than 95% of all cases [62]. It is estimated that around 200 live births were achieved in 2020 from the transplanted ovarian tissue cryopreserved for FP purposes. However, only two cases of live births have been reported in patients who underwent OTC before menarche [70,79,80]. Therefore, most experts agree that procedures should be performed within the clinical studies protocols, and outcomes should be carefully followed.

Although these procedures offer cancer patients the chance of retaining fertility [64,65], counseling is needed to explain the actual odds of both achieving a live birth and possible complications, taking into consideration the patient’s overall health, age, and type of malignancy. The meta-analysis of 34 studies published in 2023 revealed interesting results, which can significantly help during patient counseling and decision-making [81]. The authors found that the live birth rates (LBR) after IVF in patients after cancer treatment or hematopoietic stem cell transplantation were 41%, 32%, and 19% for embryo cryo-preservation, oocyte vitrification and OTC, respectively. Interestingly, spontaneous LBR after OTC was reached 33% [81]. In concordance with those results, in the study of 285 women from 5 leading European centers, LBR after ovarian tissue transplantation and spontaneous conception was higher (30%) than after IVF (21%) [68,81].

Cancer patients who receive OTC and OTT face additional obstetric challenges that increase the risk of miscarriage, such as radiation to the pelvis, uterine surgery, or surgery to the pelvis causing unfavorable conditions for OTT.

Other challenges connected with OTC and OTT include previous chemotherapy and the risks of reintroducing malignant cells together with frozen-thawed ovarian tissue [79,82,83,84]. While it was suggested that OTC should be performed before chemotherapy starts [21,79], a study performed by Dolmans et al. found that chemotherapy before OTC did not alter the results of OTT, and the authors suggested that it should no longer be considered a contraindication. This finding is also consistent with recent ESHRE recommendations [3,79]. Similarly, it was suggested not to proceed with OTC in cases of systemic diseases such as leukemia, neuroblastoma, and Burkitt’s lymphoma. However, studies that investigated the risk of relapse after OTT suggest that harvesting ovarian tissue while patients are in complete remission may be safer, especially since neither graft follicle density nor reproductive performance is significantly affected by chemotherapy administered before OTC [79,85,86,87,88]. Dolmans et al. observed a relapse in only 4.2% of cases, consistent with the data provided by Andersen et al. (3.9%) [79]. In both studies, all the relapses were dependent on the primary disease and were unrelated to OTT, as they were distant from the grafting site, and most were very close to the location of the primary cancer. These data are of utmost importance with regard to pediatric patients who often suffer from hematological malignancies, display a rapid progression of the disease, or are admitted for treatment in a severe condition. With regard to these new data, postponing OTC till remission is obtained becomes a reasonable option for such patients [79,89]. 

To avoid the relapse risk, attention has been paid to the possibility of transferring non-growing follicles to artificial scaffolds, such as 3D-printed polymer matrices, fibrin clots, and even reproductive cell-free ovaries growth [90,91]. Research is also underway on human in vitro models using stem cells as a means of securing fertility [82]. 

The thorough and personalized information about the patient’s infertility risk connected with oncological treatment should be the first and indispensable step in FP care. This is in accordance with the recommendations of the International Late Effect of Childhood Cancer Guideline Harmonization Group and guidelines from other medical societies [62,63,73,92]. The choice of the given FP method depends on many factors and has to be made on a case-by-case basis, preferably by a multidisciplinary team. Although recommendations were developed to aid the decision process [27,35,62,63,73], the decision-making process is rarely straightforward, and an individualized approach is always needed for the best FP care. The factors that need to be weighed include the patient’s pubertal status, age, risk of gonadal failure after treatment, the possibility of treatment delay, cultural or religious issues, healthcare systems and legal regulations. 

Staging the risk of gonadal failure after cancer treatment is one of the pivotal issues, as it is of utmost importance to determine whether the risks of infertility for a given patient are significant enough to justify intervention and that the benefits of preserving fertility for future use outweigh the risks of the procedures and delay of oncological treatment.

This is especially challenging, considering the quickly emerging new cancer treatments, including targeted and biological therapies and the fact that some FP procedures are still experimental, especially in prepubertal patients. The authors of the widely used Edinburgh criteria, who considered the risk of gonadal failure at above 50% as a necessary qualification criterion, quite vaguely described the process of risk estimation as based on the “relevant scientific literature and author’s own experience”. Fortunately, few recommendations have been recently published on how to approach this issue, including papers from the International Late Effect of Childhood Cancer Guideline Harmonization Group, the consensus of the Pediatric Diseases Working Party of the EBMT, the International BFM Study Group, and the Pediatric Initiative Network Risk Stratification System [42,93]. The content of this section is summarised in Table 1.

### 3.2. Oncofertility and Its Funding around the World

Approaches to FP, its availability to patients, and its legal and financial aspects differ between countries worldwide [94,95].

The United Kingdom (UK) is one of the European countries with well-developed oncofertility strategies in pediatric and adolescent populations. The UK national guidelines were developed by the Children’s Cancer and Leukemia Group and the British Fertility Society (BFS) [96,97]. Widely accepted as the primary method among sexually mature girls, embryo cryopreservation accounts for the largest percentage of oncofertility services in the UK. Cryopreservation of mature oocytes is used as an alternative to embryo freezing. In most cases, funding for this method comes from public funds from the National Health Service (NHS), the Clinical Commissioning Groups (CCGs), and the Local Commissioning Groups (LCGs). Charitable foundations are also involved in the funding process [98]. Among the methods still considered experimental are OTC in prepubertal patients and in vitro maturation [97]. The latter method is not subject to public funding due to its questionable efficacy [99]. Cryopreservation of male gametes is the primary method used in sexually mature adolescents, and the entire procedure is publicly funded by the NHS [99,100]. Cryopreservation of testicular tissue has been recently introduced in the UK and is still considered experimental [101]. Funding for tissue cryopreservation, both ovarian and testicular, mainly comes from charity, although there have also been cases of public funding as part of non-commercial research [99].

Sweden has one of the most thoroughly developed national fertility care programs for minors undergoing cancer treatment. Swedish studies have indicated that future parenthood is important to childhood cancer survivors [102,103]. The available analyses from Sweden show that doctors were far more likely to discuss the topic with patients. [102,103]. As early as 2012, the Swedish Association of Local Authorities and Regions reported the need for national guidelines on oncofertility to the Swedish Government. These were eventually released in their final version in 2015 as an annually updated guide for healthcare system professionals, published on the Swedish Human Tissue Authority website [104,105]. Preserving fertility was also included in the National Program on follow-up after childhood cancer [104,105,106]. It is also noteworthy that all Swedish university teaching hospitals have developed FP programs. Methods considered experimental include cryopreservation of ovarian tissue and cryopreservation of prepubertal testicular tissue. Therefore, these procedures are only possible in centers with Ethics Review Board approval as part of scientific research. Karolinska University Hospital has frozen ovarian tissue from 250 patients, 100 of whom were younger than 17 at the time of collection [105]. Other methods, such as cryopreservation of sperms, oocytes, or embryos, are widely accepted as a publicly funded standard [92]. Other Nordic countries where oncofertility procedures are performed include Norway, Denmark, and Finland. However, legal issues pertaining to FP procedures in minors are not established in those countries [106].

France is one of the few EU Member States to have developed FP structures dedicated to minors. In one survey of pediatric oncologists and hematologists, as many as 98% of centers selected from the French territory admitted to offering ovarian tissue cryopreservation to girls. This procedure is also funded by public healthcare [107]. Since 2004, France has had an obligation to provide FP through the current French National Cancer Plan and the Bioethics Act [108]. The legislation clearly states that anyone at risk of premature fertility change, regardless of age, has the right to benefit from its protection in the form of storage of their gametes or embryonic tissue (Art. L. 2141-11), and the method of choice for sexually mature adolescent girls is OC [109,110,111]. Funding covers the procedure for ovarian tissue collection, transplantation later in life, and in vitro fertilization but does not include annual storage fees for biological material. In boys, the most recognized method is sperm cryopreservation, although this is a problematic procedure at the onset of sexual maturation [111]. In addition, the 2022 report on the causes of infertility and the national strategic direction for combating infertility states that adolescents between 13 and 18 years of age should be entitled to ongoing, long-term fertility medical consultations and the opportunity to participate in clinical trials [112].

In the United States, oncofertility procedures are addressed in the American Society of Clinical Oncology (ASCO) guidelines. Cryopreservation of gametes and embryos is a standard and widely available FP choice for adolescents. Cryopreservation of ovarian and testicular tissues for sexually immature children is not routinely recommended, and such techniques are offered to patients only in research settings [92,113,114]. In addition to the freezing of gonadal tissues, there are two other experimental methods, i.e., gonadal suppression with GnRH analogs and in vitro maturation of oocytes or ovarian follicles [115]. The main sources of funding for FP techniques are the patient’s insurance and private funds. In 2018, the United States enacted legislation mandating coverage of oncofertility services, and thus, insurance reduced costs for some patients. Even before the legislation, some private insurers offered coverage for OC, while patients were often eligible for a discount under the Livestrong Fertility Program when choosing this technique [116]. However, coverage for fertility preservation procedures is high and can be a problem for many people, particularly if they do not have health insurance. Only a few states have enacted laws requiring private insurers to fund FP coverage for iatrogenic infertility [117]. States with FP coverage include California, Utah, Colorado, Illinois, New York, Maine, Connecticut, New Hampshire, Rhode Island, New Jersey, Delaware, and Maryland. In contrast, the states of Hawaii, Pennsylvania, and Massachusetts remain in active legislation [118,119].

In Australia, as in most parts of the world, the cryopreservation of gametes or embryos is the primary FP technique for adolescents. The cryopreservation of ovarian tissue is still considered an experimental method, while the freezing of testicular tissue is only possible in the context of clinical trials approved by the Human Research Ethics Committee (HREC) [120,121,122,123,124]. The administration of GnRH analogs to protect reproductive organs in adolescents is also carried out in the context of clinical trials, so funding comes from private sources or the resources of the hospital performing the procedure [124]. Funding for FP techniques is limited and depends on the site of treatment and the type of insurance [125]. Table 2 provides a summary of aspects related to the funding of oncofertility procedures in each country.

In the authors’ country, i.e., the Republic of Poland, embryo freezing, as well as oocyte and sperm cryo-preservation, are available to patients undergoing gonadotoxic treatment, but they are not publicly funded. Updated recommendations regarding FP in adult patients treated for malignant diseases were developed by the Polish Society of Gynecological Oncology in 2021 and agreed with European and international guidelines, including OTC not being an experimental procedure. Unfortunately, recommendations regarding OTC/OTT cannot be utilized due to the lack of necessary legislation pertaining to the harvesting, cryopreservation, and transplantation of reproductive tract tissues, including gonadal tissue. Reproductive tract organs are specifically excluded from the 2016 Polish Transplantation Act. Therefore, at present, no legal basis exists for the harvesting and transplantation of human gonadal tissues. Scholars and clinicians, including our team, have tried to change this unfavorable legal situation [40,62,63,126].

### 3.3. Oncofertility from the Perspective of Healthcare Professionals

Following a new cancer diagnosis, patients have a short time to make important decisions, including those regarding FP. The gonadotoxic potential of chemotherapy and radiation therapy varies depending on the type, dose, agent, and site of irradiation [88,127,128,129].

According to the guidelines of many organizations, all patients, regardless of risk, should receive clear and objective information about their fertility, and this should be done after the diagnosis but before starting treatment [59,92,130,131,132,133]. It is important to provide this information to every patient and those at risk to enable them to make decisions regarding FP while giving them sufficient time to comprehend the information and ask questions. Discussions about fertility risks should begin immediately after the diagnosis, ensuring that all participants understand the information provided [25,134,135,136,137,138]. Medical interviews should occur with healthcare professionals in a comfortable environment, considering the patient’s age, developmental level, cognitive functioning, and emotional maturity. It is essential for the dialogue to be empathetic and open, especially when discussing sensitive areas, such as sexual practices [88,139,140,141,142]. When informing patients, healthcare professionals should present all possible FP options while avoiding encouragement or discouragement. By providing basic information about reproductive health, they help patients understand their situation. If the patient wants more information on FP, they should be referred to specialists [132,134,143,144,145,146,147,148].

According to some studies, there is an increasing trend of multiple discussions on oncofertility, which refers to the fertility of cancer patients, to enable teenagers to understand and adapt to potential changes. This means that regular discussions take place before the start of treatment, during therapy, and after its completion [134,142,146,147,148,149]. Studies have also shown that choosing the timing of ovarian preservation in patients with large abdominal tumors, such as neuroblastoma, 3–6 weeks after treatment can offer surgical and safety benefits, despite the gonadotoxic impact after 1 or 2 cycles of chemotherapy [127]. In younger patients, especially before reaching puberty, most FP procedures are still in an experimental phase, which can make healthcare professionals hesitant to initiate discussions on this topic due to uncertainty about the available options. Fertility preservation is a sensitive subject to discuss with this patient group, as it involves issues such as body changes and sexual practices (e.g., masturbation and sexual activity) and the necessity to consider the patient’s level of sexual maturity [133,134,144,149].

During patient interviews, clinicians encounter various communication barriers that can hinder discussions about FP options [141,144,150,151,152,153]. Particularly with young patients, healthcare workers are often hesitant to address topics related to sexual practices, suggest FP methods, or discuss future family plans due to their inappropriate age or sexual maturity [131,154,155,156]. Recent studies have revealed that healthcare providers have uncertainties about how to conduct discussions about fertility preservation with young patients who should be involved in these conversations and when it is best to initiate them [157]. There is also a significant knowledge gap among medical staff regarding FP procedures, available guidelines, their effectiveness, costs, and accessible educational materials, which sometimes leads to a lack of accuracy and comprehensiveness in these discussions [135,150,151,158,159,160]. Furthermore, research indicates differences in specialists’ knowledge regarding FP procedures based on gender, noting that knowledge about the options available to girls and young women is less widespread [142,154].

### 3.4. Oncofertility from the Perspective of an Adolescent

The existing communication barriers between healthcare professionals and young patients have a significant impact on how these patients perceive information about FP. The patients often feel poorly informed, considering the information provided incomplete, confusing, and not fully understanding the risks associated with the procedures [84,130,142,145,161,162]. As a result, many of them seek information about FP methods from various sources, including different members of the medical team such as nurses, the Internet, parents, or printed materials [34,163,164]. However, after obtaining basic information, these patients strongly prefer face-to-face conversations to receive personalized information and advice [138]. It is crucial that they have access to various options for information and support, allowing them to participate in the decision-making process regarding FP actively and providing them with a sense of control over this aspect of their healthcare [138].

Moreover, it is worth noting that research focusing on adolescents has shown some gender differences in approaches to fertility discussions [130,133,165]. Young women receive insufficient information from their doctors, while young men more often remember these conversations and are more satisfied with how they were conducted [130,144,151,154,166]. Additional studies on adolescents and young adults have indicated that male patients are more inclined to discuss FP options and are more often referred to specialists in this field [137,167,168]. Despite the overwhelming challenges brought on by the diagnosis, young patients consistently appreciate the time and dedication that medical staff invest in conversations and support. They feel grateful for the honest information about their health status and value the sensitivity and commitment of medical personnel in conducting these delicate discussions [169,170]. Analyzing the literature on FP, various factors influencing the patient’s decision-making are highlighted [171,172]. One key factor is the patient’s age–research indicates that younger patients below the age of 12 often do not fully comprehend the information conveyed, whereas older patients in the age range of 12–18 may experience difficulties related to FP procedures due to their overall health condition, the lack of experience with topics such as masturbation, and the associated taboos or stigmas [173]. Additionally, other factors influencing decision-making include information about preventive measures, a fear of cancer recurrence, concerns about cancer affecting future offspring, worries about fertility behavior, and parental recommendations [59,171,172,174,175].

### 3.5. Oncofertility from the Perspective of Parents

The literature highlights divergent priorities between teenagers and their parents. Many studies indicate that parents focus on their child’s treatment and survival. As a result, they may delay discussing FP issues with teenagers, leading them to overlook the concerns of the young individuals. Parents often lack full awareness of their child’s future parenting plans, conceal their views on FP and frequently delay in deciding on this matter [25,140,176,177]. Healthcare professionals should be aware that patients may want to have children in the future. The best way to protect their fertility is to include parents in the discussion, which will promote informed decision-making about their children’s reproductive future [152,169,178,179].

The level of parental involvement varies most with the patient’s age and level of autonomy, while religion, ethical issues, and nationality have little impact [138,154,177,180]. However, a study conducted in Lebanon found that ethical, social, and religious barriers may influence FP decision-making. Other studies have shown that parents significantly influence FP decision-making, including parents’ hopes of having grandchildren [172,181]. Difficulties in decision-making are exacerbated by a lack of knowledge about fertility preservation, a lack of knowledge about the benefits, uncertainty about one’s values and preferences, and inadequate support. Therefore, physicians need to provide parents with timely, understandable, and accurate information on this topic. Studies have shown that parents wished for an exhaustive explanation of their child’s condition and treatment options, which is a common way to reduce the sense of uncertainty. Because of being overwhelmed initially by the amount of information given at the time of diagnosis, parents and caregivers are often unable to make a fertility-related decision [25,139,152,182,183]. In addition, some parents do not involve their children in the decision-making process, as their primary objective is to protect them as much as possible. So, they do not provide them with information, do not support discussing fertility problems, or want to take control of the conversation [136,140,177,184,185,186,187,188,189,190,191,192,193,194,195,196].

Research has shown that patients express a willingness to make decisions jointly with their parents and a desire to have an influence on the decisions made [136,176,177,185,197]. Although many desire parental involvement, parents usually take a greater level of decision-making involvement than the patients would prefer [178,179,185,187]. Figure 2 shows the most important communication issues between the therapeutic team and the actors involved in oncofertility procedures.

### 3.6. Legal and Ethical Aspects of Oncofertility

It seems possible that FP in children and adolescents with cancer can cause a lot of controversy in the work of a doctor. Individual countries are trying to regulate this issue, looking for the best solutions. The International Guideline Harmonization Group recommends that all patients be informed of the risks associated with the planned treatment. Advice should be given, especially to those in whom gonadotoxic treatment is planned. According to these recommendations, sperm cryopreservation is recommended for pubertal and post-pubertal patients, and oocyte or embryo cryopreservation for post-pubertal patients. The collection of ovarian and testicular tissue before puberty is moderately recommended for clinical trials only [189,190,191]. In France, there are several centers authorized to perform FP procedures. In the UK, centers performing oocyte retrieval procedures on patients under 18 require special scrutiny by the Care Quality Commission [92,105,110,111,188,192,193,194].

The clinical condition and ethical considerations are considered. In the case of invasive procedures, consideration should be given to the impact of such a procedure on the patient’s condition, the risk of the presence of tumor cells in the harvested tissue fragment, the experimental nature of some procedures, and limited data on the possibility of success of subsequent reproduction [27,105,107,111,116,188,195,196,197]. Countries with clearly developed regulations governing the legal and ethical aspects of FP that are available to authors are Sweden, France, and the UK.

In Sweden, full information should be provided to minors when they reach the age of maturity. The decision to initiate a specific procedure is made after a multidisciplinary consultation with the patients concerned and their parents—the decision and responsibility for the procedure rests with pediatric oncologists. In Sweden, there are no time limits regarding the storage of sperm and oocytes. Each time, the patient is individually informed about the storage period of the biological material. In the event of the child’s death, tissues and cells are not used for reproduction by any third parties, and they are destroyed. The patient can donate biological material for research and other medical purposes [105].

In France, the 2004 bioethics law (Art. L.2141-11 of 6 August 2004) requires FP to be offered to cancer patients. To initiate an FP procedure, consent is required from at least one parent and the adolescent concerned, in which case the objection of the other parent is ignored. For patients who cannot consent to the procedures themselves (for age-related or other reasons), the consent of one legal guardian is sufficient. An administrative route is also possible. The doctors in charge of cancer treatment refer the patient to the FP team as soon as possible (in emergency cases, within 48 h throughout the year). In the event of the patient’s death, the harvested tissue or embryonic cell material may not be used for any other purpose than originally specified [110,198].

According to the 2013 National Institute for Health and Care Excellence (NICE) recommendations, no lower age limit is set regarding the possibility of FP in oncology patients [192]. With regard to FP in minors, the UK General Medical Council recommends that it is important to check whether the young person can consent to and understands the treatment options proposed to them before making a decision. It is usually necessary to have the consent of one parent [199]. With parental consent, the collected material may be used for a research project. On reaching 16, a teenager may expressly object to the storage of gametes. The gametes of a minor may be stored without consent (also without parental consent) in justified cases certified by the doctor, in the minor’s best interest. In this case, if gametes have been stored without consent, they may not be stored after the patient’s death. The gametes of a person who has died may not be used without written consent [200,201,202]. As amended, the Human Fertilization and Embryology Act of 1990 provides for the storage of gametes and embryos for a basic storage period of 10 years (a maximum of 55 years). After this time, they must be destroyed [203,204,205,206,207,208]. Table 3 presents a summary of ethical and legal aspects in each of the countries analyzed.

## 4. Summary

Oncofertility, in terms of age-appropriate sexual development, is an effective strategy to reduce the adverse effects of anticancer therapy on the reproductive health of pediatric and adolescent populations. Efforts to disseminate and systemically pay for FP procedures in this age group are urgently needed, especially in response to the rapidly increasing survival rates among pediatric oncology patients in recent years.

Based on this review, there has been a breakthrough in oncofertility in the pediatric and adolescent populations over the past seventeen years. The use of mature OC for girls, sperm cryopreservation for boys, and embryo cryopreservation for both sexes has been approved in the populations in question. Furthermore, contemporary researchers are seeking to develop methods that can be used for younger age groups, especially the Prepubertal ones. Many countries have decided to fully fund or publicly co-fund FP procedures for patients before age 18 who receive gonadotoxic therapy. Unfortunately, this is still a small number, as most governments do not choose to do so and do not even regulate the need for information on the possibility of preserving reproductive function before the use of gonadotoxic therapy. Many organizations such as ASCO, ASRM, NICE, and SIOPE call for the promotion of FP in young cancer patients. However, there are several barriers to the widespread implementation of oncofertility, including, but not limited to, the knowledge and experience of healthcare professionals and ethical or worldview aspects of both parents and patients. Suppose the approach to payment and financing is changed in the future. In that case, appropriate educational measures for the public and medical professionals are implemented, FP procedures are made more widespread, and the patients’ quality of life will likely improve. In addition, appropriate legislation may be required in many countries to reduce barriers to effective and sufficient funding and payment for oncofertility.

This article may promote the implementation and effectiveness of research, especially in real-world settings, to describe systemic solutions for FP methods among pediatric and adolescent patients more broadly. Without the appropriate knowledge on this topic and experience of change from others, it will be difficult to encourage other state payers to implement systemic solutions for oncofertility in this age group. It is also crucial to better explore the perspective of patients and their parents.

## 5. Limitations

The review conducted is subject to several limitations. The first is that it was not possible to reach and include in the analysis all available studies on FP in the pediatric population. On top of that, a certain degree of subjectivity in including records in the review cannot be ruled out. Admittedly, the data were extracted by a single researcher, and a thorough pre-publication check was performed, but double-independent extraction was not used. Any doubts about the inclusion were discussed by the entire team of authors. We did not attempt to contact authors of articles we could not retrieve.

The synthesis of the methodological features of the review prevents a full qualitative assessment of the studies. This implies that the review is only as good as the reviews and studies included. It is also possible that the findings presented in this publication reflect some methodological or conceptual errors in the included studies.

An attempt was made to select the optimal interval for the inclusion of primary studies, but it is not as wide as for systematic reviews, which means that they may not represent a comprehensive and multi-year cross-section of available studies.

## 6. Conclusions

During cancer treatment of pediatric and adolescent patients, attention should be paid to patient education regarding the understanding of the impact of treatment on fertility. In discussions about the quality of pediatric oncology care, the process of introducing national standards for safeguarding the fertility of this population should be initiated, exemplified by the countries outlined in the manuscript. This manuscript can help in the development and introduction of certain systemic solutions. It should be emphasized that FP should be an option for patients, regardless of age and financial resources.

## Figures and Tables

**Figure 1 cancers-15-04393-f001:**
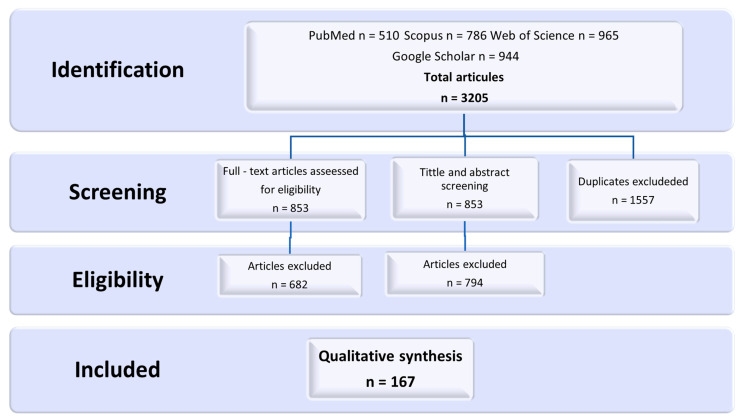
A flowchart of the inclusion and exclusion process.

**Figure 2 cancers-15-04393-f002:**
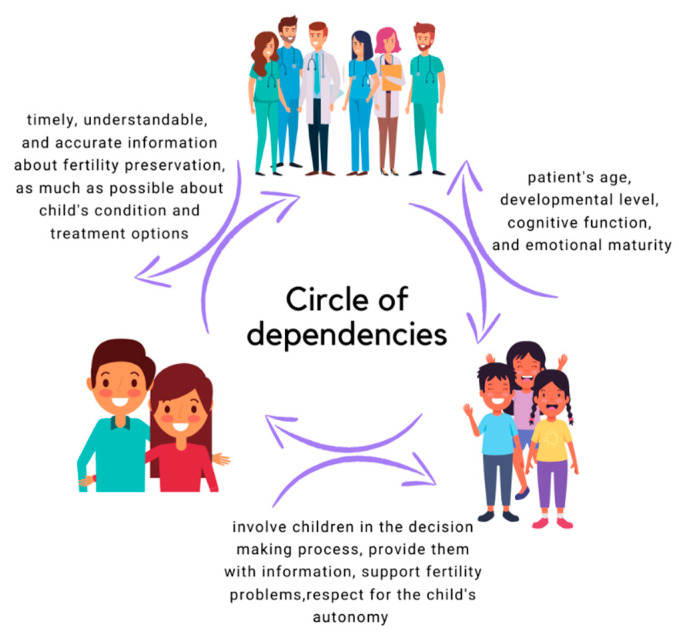
A circle of communication dependency between children and adolescents, parents, and medical staff.

**Table 1 cancers-15-04393-t001:** Oncofertility methods used in pediatric and adolescent patients.

Method	Puberty	Legislation	References
Female Patients			[41,42,43,44,45,46]
Embryo cryopreservation	Postpubertal	Established	
Mature oocyte cryopreservation	Postpubertal	Established	
In vitro maturation	Prepubertal Postpubertal	Experimental	
Ovarian tissue cryopreservation	Prepubertal Postpubertal	Experimental	
GnRH analogs	Postpubertal	Experimental	
Male Patients			[42,43,44,45]
Embryo cryopreservation	Postpubertal	Established	
Sperm cryopreservation	Postpubertal	Established	
Testicular tissue cryopreservation	Prepubertal Postpubertal	Experimental	

**Table 2 cancers-15-04393-t002:** Funding of children and adolescent FPmethods in selected countries in the world.

Country	Method	Funding	Additional Information
UK	Cryopreservation of embryos and oocytes. Cryopreservation of ovarian tissue and testicular tissue in underage patients.	Public financing from the NHS, CCGs, LCGs, and foundations.	The cryopreservation of embryos is a widely accepted method for sexually mature girls. Funding often comes from public funds of the NHS, CCGs, LCGs, and charitable initiatives.
Sweden	Cryopreservation of embryos, oocytes, ovarian tissue, and testicular tissue.	Charitable funds and grants.	Sweden has an extensively developed fertility care program for patients undergoing oncological treatment. All Swedish university hospitals have established fertility care programs. The cryopreservation of ovarian tissue is conducted in centers approved by the Ethics Review Board as part of scientific research.
France	Cryopreservation of ovarian and testicular tissue for underage patients.	Healthcare funding, a standard for adolescent patients.	France has introduced the obligation to provide oncofertility through the current National Cancer Plan and the Bioethics Law. Funding includes tissue collection, future transplants, and in vitro fertilization. France has implemented specific legislation requiring oncofertility provision for patients at risk of fertility loss.
USA	Cryopreservation of gametes and embryos is a standard for adult patients. Experimental methods are only applied in research.	Patient’s insurance and private funds. The 2018 law introduced the obligation for insurance coverage for oncofertility services.	The use of GnRH analogs and in vitro maturation is still experimental. Various states in the USA have different regulations concerning insurance coverage for oncofertility.
Australia	Cryopreservation of gametes and embryos is the standard for adolescent patients.	Limited funding is dependent on the location and type of insurance.	The cryopreservation of ovarian tissue is considered an experimental method, and testicular tissue is cryopreserved only within research approved by the Ethics Review Committee.

**Table 3 cancers-15-04393-t003:** A summary of ethical and legal aspects in each of the countries analyzed.

Country	Legal and Ethical Aspects	Year of Publication of the Source Materials
Sweden	Consent is required from the patient and both parents. The decision and responsibility for the procedure lies with the attending physician. No time limits apply to the storage of sperm and oocytes. In the event of the child’s death, the tissues and cells are destroyed There is a possibility of donating biological material for research and other medical purposes (with parental consent).	2023
France	Consent is required from the minor and at least one parent (an objection of the other parent is ignored); an administrative route is also possible. The decision and responsibility for the procedure lies with the attending physician. FP consultation should occur as soon as possible (in emergencies, within 48 h throughout the year). In the event of the child’s death, the tissues and cells may not be used for any other purpose.	2004, 2022 and 2023
United Kingdom	No lower age limit is set as to the possibility of FP. Before deciding, it must be established whether the patient is mature enough to understand the FP procedure and can make an informed decision. Consent is required from the patient and one of the parents. On reaching 16, a teenager may expressly object to the storage of gametes. Gametes may be stored without the patient’s and parental consent in justified cases. There is a possibility of donating biological material for research and other medical purposes (with parental consent). The maximum storage period of gametes and embryos is 55 years. The basic storage period is ten years and can be extended for legitimate medical reasons (a maximum of 55 years). After this time, they must be destroyed.	2013 and 2023

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
