# Peer review of "Fertility Preservation in Children and Adolescents during Oncological Treatment—A Review of Healthcare System Factors and Attitudes of Patients and Their Caregivers"

_cancers, 2023, doi:10.3390/cancers15174393_

Round 1

Reviewer 1 Report

The manuscript concerns the problem of oncofertility, but only takes into account selected aspects, such as methods of fertility preservation.The title is too general, it does not indicate what problems will be discussed. The subtitle "oncofertility methods" is in my opinion not very correct, I would suggest using "fertility preservation”. I propose to specify in the introduction to which group of patients the problem of fertility preservation concerns and whether it is similarly solved in European countries and the US. I think this chapter(Oncofertility methods)   is too long; it would be better to analyze the methods of counseling on fertility damage (when, who, system of informing patients and parents).

In the part „Oncofertility and its funding around the world” the authors describe the procedures in selected European countries, the US and Australia. Why were these countries chosen? The authors come from Poland - how the problem is solved in Poland, in Central Europe. I would suggest presenting this part in the form of a table, indicating the year from which the information comes. Figure 2 is  incomprehensible!- no information on the world map. A map of the world and only single countries described suggest that oncofertility program exists only in those countries.

Similarly, in the chapter " Legal and ethical aspects of oncofertility” the differences between countries should be presented in the form of a table .

I would suggest to analyze additionally:

- how the program works in individual countries in relation to the recommendations of International Late Effect of Childhood Cancer Guideline Harmonization Group

-the difficulties in fertility preservation procedures, especially in the youngest patients, especially in the case of rapid progression of the disease, severe condition of the patient

Author Response

Thank you. 

Reviewer 2 Report

This review is comprehensive and represents a valuable contribution to current knowledge. Very little is currently reported in the literature about the impact and late effects of childhood fertility preservation following cancer diagnosis, although this is now standard practice is most developed countries including the UK and Sweden. 

The introduction is succinct and informative. 
The materials and method section - I think the purpose of the review in relation to the research questions could be clearer. 
Results is a little long and would be improved by being made shorter and improving the referencing. There are many instances of statements which are not supported by a reference. Finally, the word "subjected' should be changed to something less emotive, e.g. 'underwent or undertook' (line 157).
Section 3.2 Oncofertility and its funding around the world, again is very lengthy and much has been reported elsewhere already. An overview of the practices in each country could be given, perhaps, in tabular form to help reduce the word count. 
Line 279 sentence needs completing. Line 282 the reference number 109 is not formatted in the references at the end. Line 312 "as mentioned in the paragraph above" - it is not clear what is being referred to here. The map (line 147-348) is not helpful I don't think because it doesn't give much information and takes up a lot of space. I am not sure it adds value to the review. The use of flags and male/female symbols also seems a little odd - need to consider non-binary people and gender as sensitive topics in this field. Not all flags will be familiar to readers. Again, using a tabular form to present results may work better for this section. 

Section 3.3. Oncofertility from the perspective of adolescent patients. This section is rather long and some of it is rather repetitive - second paragraph particularly. This whole section could be shorter and more succinct. The referencing is not properly done with many statements made without references.  E.g. line 403 refers to interviews, but as no reference is given, the reader does not know which interviews are being referred to. Overall, this section is hard to navigate as a reader and get a sense of the main points in relation to the aim and the research question. I think this section needs the most revision.

Section 3.4 line 422 statement that parents may not notice the young persons fertility-related concerns needs referencing - this is not an accurate blanket statement and has not been widely accepted as opinion. Line 437 sentence ending in 'possible' on line 438, this needs referencing for the same reason as the previous comment about line 422. Line 450 there are too many references grouped together, it is hard to decipher the significance of the statement, again this could possibly be added in tabular form if the information is essential. If this point is going to be made I think it needs to be made properly - describe what the legislation is and how it differs, if this is important. But this section is about the perspective of parents so this point may be better elsewhere e.g. in legal and ethical aspects. 

Figure 3. This is cute.

Section 3.5 - again a little longer than needed. Also needs a summary and improved structure, it's not entirely clear what the most important findings are here.

Summary - it's OK, could come together better after editing the rest, to highlight the key points in relation to the research question/aim. Line 578 has a \ typo at the end of the sentence.  

Other: I think a section on limitations would be relevant. 

See examples as given above. As a whole, the article could do with some revision in this area - due to the writers not having english as their first language. They have done a good job but some language is unclear in places. See lines 511 and 512 as example. 'Each time the consent of the minor patient is considered and indicated for the initiation of the procedure when the patient is an adolescent." - potentially not clear, could be revised to: "Each time the procedure is initiated for an adolescent patient, their consent is carefully considered and obtained:. 

Author Response

Thank you. 

Round 2

Reviewer 1 Report

The changes made to the manuscript respond to my comments, sometimes - the text is overly detailed, such as the technical description of the procedures in females/ males of fertility preservation (FP) as well as  "oncofertility from the perspective of healthcare professionals/ parents and patients" . These text fragments could be shortened.The authors still do not refer to the possibilities of FP in other European countries, including Poland - are there no FP procedures there?

Author Response

Thank You.
